# The Role of Gut Microbiota in an Ischemic Stroke

**DOI:** 10.3390/ijms22020915

**Published:** 2021-01-18

**Authors:** Ryszard Pluta, Sławomir Januszewski, Stanisław J. Czuczwar

**Affiliations:** 1Laboratory of Ischemic and Neurodegenerative Brain Research, Mossakowski Medical Research Institute, Polish Academy of Sciences, 02-106 Warsaw, Poland; sjanuszewski@imdik.pan.pl; 2Department of Pathophysiology, Medical University of Lublin, 20-090 Lublin, Poland; czuczwarsj@yahoo.com

**Keywords:** brain ischemia, stroke, gut microbiota, amyloid, tau protein, trimethylamine-n-oxide

## Abstract

The intestinal microbiome, the largest reservoir of microorganisms in the human body, plays an important role in neurological development and aging as well as in brain disorders such as an ischemic stroke. Increasing knowledge about mediators and triggered pathways has contributed to a better understanding of the interaction between the gut-brain axis and the brain-gut axis. Intestinal bacteria produce neuroactive compounds and can modulate neuronal function, which affects behavior after an ischemic stroke. In addition, intestinal microorganisms affect host metabolism and immune status, which in turn affects the neuronal network in the ischemic brain. Here we discuss the latest results of animal and human research on two-way communication along the gut-brain axis in an ischemic stroke. Moreover, several reports have revealed the impact of an ischemic stroke on gut dysfunction and intestinal dysbiosis, highlighting the delicate play between the brain, intestines and microbiome after this acute brain injury. Despite our growing knowledge of intestinal microflora in shaping brain health, host metabolism, the immune system and disease progression, its therapeutic options in an ischemic stroke have not yet been fully utilized. This review shows the role of the gut microflora-brain axis in an ischemic stroke and assesses the potential role of intestinal microflora in the onset, progression and recovery post-stroke.

## 1. Introduction

A brain ischemia results in brain injury caused by transient or permanent or focal or global stop of the cerebral blood flow that can lead to permanent neurological deficits, dementia or death [1,2]. A brain ischemia in people called a stroke is a global health problem that has now become the second leading cause of death and the third most common cause of disability worldwide [3]. In humans, a stroke is classified as ischemic or hemorrhagic based on the underlying neuropathology [3]. Ischemic strokes account for 85% of all cases and hemorrhagic strokes for about 15% [3]. An ischemic stroke is mainly caused by occlusion of the middle cerebral artery, which causes damage to the brain parenchyma in the affected region followed by a neuroinflammatory and immune response [1,4,5]. Brain damage due to an ischemic stroke is the result of a complex series of neuropathophysiological and neuropathological events including excitotoxicity, oxidative stress, neuroinflammation, apoptosis, amyloid production and tau protein dysfunction [4,5,6,7,8,9,10,11]. The post-ischemic brain is characterized by the accumulation of amyloid plaques and neurofibrillary tangles with a subsequent development of dementia [9,10,11,12,13]. It therefore will become a serious public health problem with morbidity and prevalence reaching epidemic proportions in the next few decades if the disease cannot be prevented or slowed down. Although a stroke increases neurological deficits with dementia, infections are a major cause of death from a stroke [14]. About 90% of stroke cases have been documented to be associated with behavioral factors including poor nutrition, low physical activity and smoking as well as metabolic factors including diabetes, obesity, hyperlipidemia and hypertension [15]. According to global disease research, a stroke is and will be a very serious health problem; the negative influence of which will grow with the aging of the world population [16]. In addition, there is a mechanistic relationship between cerebral ischemia, innate and adaptive immune cells and intracranial atherosclerosis as well as intestinal microflora in the modification of brain responses to an ischemic injury. Experimental studies have highlighted the cellular and tissue mechanisms associated with stroke damage; mechanisms that have identified new pathways of damage that have not yet been accurately described. After a stroke, up to 50% of patients experience gastrointestinal complications including constipation, dysphagia, gastrointestinal bleeding and stool incontinence [17,18,19,20]. Gastrointestinal complications after a stroke affect poor patient treatment results including a delayed outcome, increased mortality and progressive neurological deficits. Recently, a few studies have also shown that impaired intestinal microflora can also be a risk factor for a stroke and can affect the prognosis after a stroke [15,21]. Additionally, other studies have shown a significant influence of the gut microbiome on the pathogenesis of various cerebrovascular diseases [22]. Human intestinal microflora consists of tens of trillions of bacteria with about 1000 species of known bacteria and about three million genes, which is 150 times more than the human genome [22]. The brain and intestine are connected by a neuronal network, forming a complex gut-brain-gut axis with strong bilateral interactions. Increasing data indicate that intestinal microflora is an important factor in the development, sequelae and treatment of a stroke. An ischemic stroke also changes the composition of the intestinal microflora. Conversely, the gut microflora can modulate the outcome of a stroke and play a role in its development. From a clinical point of view, the risk of a stroke remains a huge challenge today.

It is believed that the digestive tract is the main organ of the immune response, which is rich in immune cells and responsible for more than 70% of the activity of the whole immune system [15]. Increasing evidence suggests that enteritis along with the immune response plays an important role in neuropathophysiology of a stroke, which may become an important therapeutic target in treating the consequences of a stroke [23]. The intestinal microflora has been shown to play an important role in regulating the immune system [15]. The intestinal microflora has also been shown to be an important factor in the development and sequelae of a stroke in a mouse model [24,25,26,27]. In addition, a stroke usually triggers intestinal dysfunction, intestinal microflora dysbiosis and intestinal bleeding as well as septicemia of intestinal origin, which affects a poor prognosis [15]. Stroke-induced gastrointestinal complications [17,18,19,20] coexist with poor post-stroke outcomes (for instance, increased mortality, worsening neurological functions and delayed recovery time) [15,19,21,24,28,29]. However, the less known secondary effect of a stroke is the intestine microbiota dysbiosis. It is known that imbalances in the intestinal microflora contribute to neuro-behavioral problems, neuroinflammation and to worsening stroke outcomes [28,29]. There is little information on the characteristics of the intestinal microflora from stroke patients [30]. Often, these complications adversely affect the outcomes of a stroke. Recent studies have postulated the role of the brain-gut axis in causing dysbiosis of the intestinal microflora and various complications and negative outcomes post-ischemia [31,32,33]. In this review, we present our recent understanding of the interaction between the intestinal microbiome and the brain in determining the course of an ischemic stroke and reveal related pathways that may be promising therapeutic targets. This review also discusses ongoing research into the production and role of trimethylamine-n-oxide, its association with strokes and other pathogenic stroke mechanisms and trimethylamine-n-oxide-based therapeutic strategies.

## 2. Post-Ischemic Brain versus Gut Microbiota

Studies indicate the effect of intestinal microflora on a host stroke outcome, paying attention to two-way communication along the brain-gut axis [28,34]. Growth of *Bacteroidetes* after an ischemia was confirmed in monkeys [35]. Increased *Bacteroidetes* abundance was also found three days after the occurrence of an ischemic stroke in mice, which is considered a characteristic feature of post-stroke dysbiosis [28]. In contrast, a clinical study in which stool samples were taken for two days after admission showed a decrease in *Bacteroidetes* levels in patients with an acute ischemic stroke and a transient ischemic attack [36]. In the study of monkeys after a focal brain ischemia, an increased relative abundance of *Prevotella* was found, suggesting that this type may be associated with an inflammatory response following a stroke [35]. In monkeys after a local cerebral ischemia, reduced relative levels of *Faecalibacterium*, *Streptococcus*, *Lactobacillus* and *Oscillospira* were observed [35]. *Faecalibacterium* and *Oscillospira* species are recognized as the main source of butyrate in the host body [37,38]. Butyrate, the short chain fatty acid, plays a key role in maintaining the integrity of the intestinal barrier, inhibits the production of pro-inflammatory cytokines and is considered a therapeutic target in brain disorders [39]. A decrease in plasma butyrate concentration in monkeys after a focal cerebral ischemia was observed within 6–12 months, which was probably associated with a decrease in *Faecalibacterium* and *Oscillospira* levels [35]. The reduced plasma levels of short chain fatty acids found in monkeys after a focal cerebral ischemia with a survival of 6–12 months indicated that chronic intestinal dysbiosis may also affect the production of short chain fatty acids.

It was noted that *Lactobacillus*, which is an important type of host probiotic bacteria, possessed a reduced relative level after an infarction of the brain in monkeys [35]. *Lactobacillus* supplementation has been shown to improve cognitive function, mood and alleviate aging-related inflammation [40,41,42]. Dementia and depression after a stroke are common complications in animals and survivors of a stroke [43,44,45] with simultaneous chronic systemic and cerebral inflammation [4,5]. Whether *Lactobacillus* supplementation is beneficial for post-stroke patients, the case is open and should be investigated in future clinical trials. It has also been found that the relative level of *Streptococcus* abundance is reduced after a cerebral ischemia. The genus *Streptococcus* includes probiotic bacteria such as *Streptococcus thermophilus* [46] and pathogenic bacteria such as *Streptococcus pneumoniae* [47]. At present, the exact role of intestinal *Streptococci* in cerebral infarctions remains to be clarified in future studies. An increased blood lipopolysaccharide has been found to cause brain neuroinflammation, blood-brain barrier alterations, brain edema and complicates post-stroke survival [48]. This correlates with the increase in plasma lipopolysaccharide in post-ischemic monkeys especially 6 and 12 months after a stroke [35]. Thus, lipopolysaccharide may play an important role in chronic systemic inflammation after a stroke, which is confirmed by damage to the intestinal mucosa barrier and morphological damage to the intestinal mucosa after an ischemia. The damaged intestinal mucosal barrier is probably associated with an increased release of lipopolysaccharide from the intestines into the bloodstream. It was confirmed that the pro-inflammatory cytokines IFN-g, IL-6 and TNF-α were elevated in plasma up to 12 months after a focal brain ischemia, suggesting that systemic inflammation persisted chronically after a stroke [35]. These observations suggest that not only intestinal microflora dysbiosis develops after a cerebral infarction but also chronic systemic inflammation as well. Correlation studies also revealed that increased plasma lipopolysaccharide or inflammatory cytokine levels and excessive *Bacteroidetes* growth were closely related [35]. We can conclude that the chronic systemic inflammatory response after a stroke can affect the brain because it has been proven that this inflammatory response is associated with cognitive impairment, learning and memory impairment, depression and anxiety [40]. Pro-inflammatory cytokines released from the gut into the circulation communicate directly with the brain intensifying pathological changes [40]. Thus, post-stroke intestinal microbiota and chronic systemic inflammation may be therapeutic targets in the treatment of a stroke.

The removal of intestinal bacteria by an initial antibiotic treatment has been shown to worsen the outcome of the post-ischemic mouse [28,29]. In mice, the composition of the cecum microflora was found to change following a local cerebral ischemia, identifying specific changes in *Peptococcaceae* and *Prevotellaceae* that correlated with the extent of the injury [25]. It has also been shown that dysbiosis affects the outcome of an ischemic stroke by suppressing the effector T cell movement from the intestine to leptomeninges in a model of a local cerebral ischemia [34].

A few microflora studies in stroke patients have shown gut dysbiosis associated with systemic inflammation and a reduced trimethylamine-n-oxide level [36,49,50]. An ischemic stroke in patients was associated with more opportunistic pathogens such as *Enterobacter*, *Megasphaera* and *Oscillibacter* and in minority with beneficial types such as *Bacteroides*, *Prevotella* and *Faecalibacterium* [36]. It was then found that a stroke was independently associated with an increase in *Atopobium* and *Lactobacillus ruminis* and a reduced level of *Lactobacillus sakei* [50]. In another study, stroke patients had a dysbiosis of the intestinal microflora with increased levels of *Escherichia*, *Bacteroidetes*, *Megamonas*, *Parabacteroides* and *Ruminococcus* [15]. In patients with a high risk of a stroke, the enrichment of opportunistic pathogens with reduced levels of butyrate-producing bacteria was noted [21]. A significant increase in *Odoribacter*, *Akkermansia*, *Ruminococcaceae*, *Flavobacteriaceae* and *Victivallis* was observed in stroke patients [15]. *Odoribacter*, *Akkermansia*, *Ruminococcaceae* and *Victivallis* are well known producers of short chain fatty acids including acetate, propionate and butyrate [21,51,52,53]. In patients with a mild stroke, *Enterobacter*, *Pyramidobacter* and *Lachnospiraceae* increased their level [15]. In contrast, patients with a severe stroke had an increase in the number of *Ruminococcaceae* and *Christensenellaceae* [15]. These observations suggest that gut flora is involved in the human stroke and correlates with its severity. It was unequivocally found that short chain fatty acids were produced by increased *Odoribacter* and *Akkermansia* in stroke patients [15]. Earlier studies have shown that the shift in microbiological composition caused by a stroke was associated with an increase in *Akkermansia muciniphila* and an excessive number of *Clostridial* species in post-stroke mice [27]. On the contrary, it was observed that *Akkermansia* decreased in post-stroke patients [49]. Another study found that *Akkermansia* increased significantly in post-stroke patients [15]. It has been reported that *Akkermansia muciniphila* uses mucin to produce high-level acetate that can be used by butyrate-producing *Ruminococcaceae* to stimulate butyrate production [21]. Another study showed that the type *Odoribacter*, a producer of butyrate that belongs to the type *Bacteroidetes*, also increased after a stroke in patients [15]. This indicated that the simultaneous growth of these two bacteria could promote butyrate production. Butyrate is a preferential source of energy for epithelial cells and maintains epithelial health [54]. Butyrate may also affect the expression level of genes stimulated by *Akkermansia muciniphila* in epithelial cells [55]. Therefore, some believe that *Akkermansia* may play a key role in wound healing by promoting butyrate levels, which has resulted in the fixation of epithelial integrity in mice after a stroke [27]. In addition, *Akkermansia muciniphila* can induce mucus production and Reg3γ expression in the colon, resulting in microflora remodeling [56]. Therefore, further work is needed to determine the possible role of *Akkermansia* and *Odoribacter* in post-stroke patients.

## 3. Post-Ischemic Amyloid and Tau Protein versus Gut Amyloids

Amyloid peptides or an amyloid fiber (of a curly type), associated with forming seeds for aggregation of a brain amyloid, may be produced by some *Enterobacter* species and fungi [57,58,59]. Microbial amyloids trigger the nucleation of amyloid aggregates with a huge accumulation and cause an inflammatory response [60,61]. In the absence of intestinal microflora, there was a decrease in the amyloid accumulation in transgenic mice [62]. In addition, it has been noticed that amyloid aggregation in vitro may be inhibited by the intestinal microflora-produced short chain fatty acids [63]. Subsequently, a bacterial endotoxin is responsible for neuroinflammation triggering the formation of amyloid fibrils [64,65]. Some bacteria, such as *Escherichia coli*, produce amyloid [65] but the relationship of this amyloid to neurodegeneration in the brain after an ischemia has not been clarified. It should be added that bacterial gram-negative lipopolysaccharide supports amyloid deposition in the brains of mice, harmfully affecting cognition [66]. It is not known how bacterial amyloids cooperate with other neuropathological mechanisms in the post-ischemic brain such as post-translational tau protein changes, β-amyloid peptide generation, neuroinflammation and cerebrovascular degeneracy. When the intestinal barrier permeability increases, bacterial amyloids enter the systemic circulation increasing inflammation in the brain and causing memory impairment in mice [67]. These observations indicate that bacterial amyloids may play a key role in increasing the immunoreaction and nucleation of amyloid aggregates in the brain post-ischemia. An amyloid produced by a microbiome has the potential to affect amyloidogenesis in neurodegenerative diseases [68]. These observations have recently been supported by in vitro and in vivo data [69,70]. Microbial amyloids can control brain inflammation and β-amyloid peptide levels by influencing brain gliosis [4,5,68,69,70]. The altered bacterial flora may affect the level of bacterial amyloids and metabolites in the plasma and therefore may act as a trigger in the onset and exacerbation of neurodegeneration in the post-ischemic brain.

Meta-analysis data indicate that gut metabolites may exacerbate neuroinflammation, the aggregation of amyloid and tau protein in brain neurodegenerative diseases [71]. Gut microbiota is involved in the secretion of more than 100 metabolites but their contribution to the neuropathogenesis of the post-ischemic brain has not been proven [72]. Valerian, isobutyric, isovaleric, butyric, acetic, propionic and formic acids have been shown to affect the development of neurodegenerative diseases by acting on astrocyte and microglia activity and reducing neuroinflammation, amyloid and tau protein aggregation [73,74]. Host gut microflora was found to affect microglia homeostasis in the brain, which produced apparent microglia defects with an immature phenotype in germ-free animals, eventually impairing innate immune responses [75]. An attempt to re-colonize with complex microflora partly restored the physiological characteristics of microglia [75]. In summary, this observation suggests that the gut microflora may control microglia maturation, function and activation and therefore, in the case of impaired gut microflora, microglia maturation and its potential for amyloid and tau protein phagocytosis that accumulate after ischemia are limited. In vitro studies have shown that propionic, valeric and butyric acids produce an inhibition of oligomerization of the β-amyloid-(1–40)-peptide [63]. However, when the effect of gut microbiota metabolites on the aggregation of the β-amyloid-(1–42)-peptide was assessed, it was shown that only valeric acid inhibited the formation of amyloid oligomers [63]. In contrast, a study on the conversion of β-amyloid peptides into β-amyloid peptide fibrils showed that both butyric and valeric acids inhibited the conversion of a monomer of a β-amyloid-(1–40)-peptide to a filamentous β-amyloid peptide [63]. The dependence on the type of intestinal metabolite indicated that an increase in the amount of beneficial metabolites produced by the intestinal flora and particularly anti-inflammatory bacteria may support the removal of tau protein and amyloid from the brain tissue following an ischemia.

## 4. Trimethylamine-n-oxide and Stroke

Trimethylamine-n-oxide is a by-product of the intestinal microflora closely related to a stroke [76]. In addition, trimethylamine-n-oxide is directly associated with poor treatment outcomes in ischemic brain injury patients regardless of traditional risk factors [76,77]. A limited number of studies suggest that trimethylamine-n-oxide plays a protective role but other studies suggest that trimethylamine-n-oxide is an indicator of disrupted homeostasis rather than a causative or protective factor [78,79]. There are currently relatively few studies in the literature on the relationship between trimethylamine-n-oxide and a stroke. A clinical trial in the Chinese hypertensive population revealed that elevated trimethylamine-n-oxide levels were associated with an increased risk of a first stroke [80]. In addition, patients with lowered folic acid and high trimethylamine-n-oxide had the highest frequency of strokes [80]. In patients after the first stroke, elevated levels of trimethylamine-n-oxide were associated with the risk of a recurrent stroke. This relationship persisted even after adjusting for traditional cerebrovascular risk factors and the initial severity of the stroke. Additionally, the concentration of trimethylamine-n-oxide in the serum is closely related to the number of pro-inflammatory monocytes [81]. Clinical trials in patients after a stroke and a transient ischemic attack have shown a significant dysbiosis of the intestinal microflora. Importantly, stroke and transient ischemic attack patients exhibited lower plasma trimethylamine-n-oxide concentrations when compared with control cases with asymptomatic atherosclerosis. The authors gave an explanation that they studied the trimethylamine-n-oxide level in patients who had already had a stroke or transient ischemic attack—the level of trimethylamine-n-oxide being evidently low in comparison with a former Western investigation—and the management of the stroke or transient ischemic attack might reduce the trimethylamine-n-oxide levels [36]. A multicenter study provided evidence that the serum trimethylamine-n-oxide concentration prior to carotid artery stenting was found to be significantly higher in cases exhibiting new injuries on a post-stenting diffusion-weighted image compared with patients having no new injuries. Following adjustments for possible confounding factors, increased blood trimethylamine-n-oxide levels served as a self-dependent predictor of new lesions on a diffusion-weighted image after carotid artery stenting. Furthermore, elevated plasma trimethylamine-n-oxide concentrations have been suggested to increase carotid intima-media thickness in patients with risk for diabetes type 2 regardless of insulin resistance, fatty liver and visceral obesity. When a lifestyle was sufficiently modified, the carotid intima-media thickness was significantly reduced in patients experiencing the greatest decrease (> 20%) in trimethylamine-n-oxide levels [82]. Trimethylamine-n-oxide is responsible for neuronal senescence, synapse dysfunction and reduction of the synaptic plasticity [83]. Trimethylamine-n-oxide may play a role in Alzheimer’s disease pathology as a factor promoting brain vessel diseases. As a matter of fact, vascular risk is common in Alzheimer’s disease patients and cerebrovascular events are frequently associated with Alzheimer’s disease pathology [84].

## 5. Platelets and Trimethylamine-n-oxide

Studies conducted on animal models and healthy volunteers indicated that trimethylamine-n-oxide contributes directly to platelet hyper-reactivity and increases the risk of thrombosis [22]. Platelet activation by a number of agonists providing a sub-maximal stimulus was further potentiated by direct exposure to trimethylamine-n-oxide in a process mediated by an augmented release of Ca^2+^ from intracellular stores [22]. The trimethylamine-n-oxide-mediated effect on platelet thrombosis and hyperactivity potential was also found in a microbial transplantation investigation with the use of germ-free animals [22]. These data suggest that therapies targeting trimethylamine-n-oxide could exert a desired antithrombotic activity with a risk of bleeding complications not being increased [85,86].

## 6. Conclusions

We have presented the documented role of the gut microbiome in development and recovery from an experimental cerebral ischemia and a stroke (Figure 1). While it is well known that the microbiome influences numerous metabolic and immunological aspects of a cerebral ischemia, our understanding of how exactly the microbiome modulates brain function before and after a cerebral ischemia is still limited. Research on the gut-brain axis focuses mainly on the relationship between the composition of the gut microbiome and disease progression. Although significant progress has been made over the past few years, it is clear from the review of available publications that much remains to be clarified in relation to the association of the intestinal microbiome with a stroke. However, the main obstacle to the clinical translation of microbiome study is the large variability between patients’ intestinal microbiomes, which cannot be easily reproduced with animal models. Nevertheless, microbiome-based treatments can have a huge impact on improving post-stroke outcomes in the future [34,87,88].

As already shown above, the intestinal microflora is closely related to the activity of the immune system [15] and the subsequent modulation of neuroinflammation and stroke outcomes [28,29]. A question arises on the putative mechanisms of microflora’s action. As already mentioned, microbial metabolites belonging to short chain fatty acids, acetate, butyrate and propionate affect immune cells through free fatty acid receptors. In this way, they modulate immune and inflammatory responses. Remarkably, via these receptors short chain fatty acids may regulate the sympathetic nervous system activity [89]. In addition, short chain fatty acids have been documented to inhibit histone deacetylases which, among other actions, is responsible for the reduced production of the pro-inflammatory tumor necrosis factor and decreased activity of nuclear factor kappa B (a transcription factor). Evidently, dysregulation of the intestinal microflora via the mechanisms listed above may have an influence on immune and inflammatory reactions including those observed within the central nervous system [89].

The dysregulation of intestinal microbiota is associated with inflammatory bowel disease [90]. In an experimental model of this disease in mice, an expression of colonic matrix metalloproteinase-9 was elevated and this led to alterations in the fecal microbiome [91]. There is an association between inflammatory bowel disease and strokes and the gut-brain-microbiota axis is very likely to be bi-directionally involved in this phenomenon [92]. A possibility thus arises that matrix metalloproteinase 9 may participate in the putative mechanisms of microflora.

Finally, as research into the role of the gut-brain axis in a stroke is in its infancy, bringing in broader insights of the stroke field, current rodent models and the areas that are being explored will generate holistic questions for investigators to focus on in the future.

## Figures and Tables

**Figure 1 ijms-22-00915-f001:**
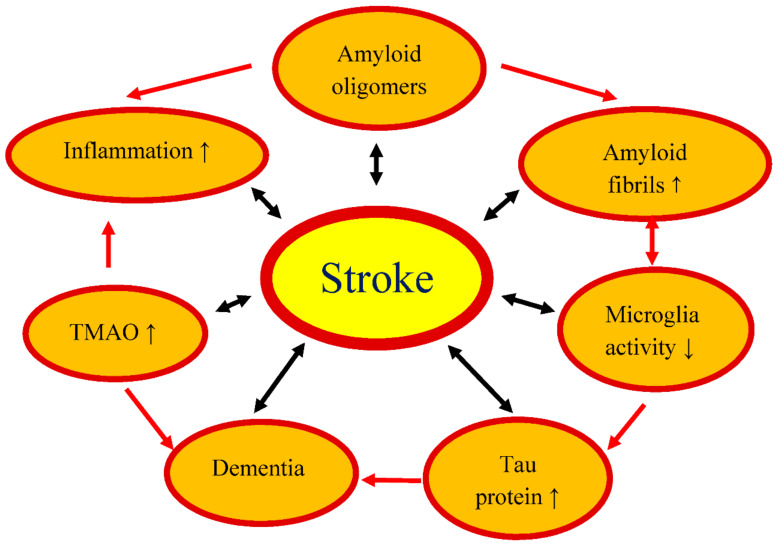
Influence of intestinal microflora dysbiosis on the development and outcome of an ischemic stroke. ↑: increase, ↓: decrease, TMAO: trimethylamine-n-oxide.

## Data Availability

Not applicable.

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
