# Peer review of "The Role of Gut Microbiota in an Ischemic Stroke"

_ijms, 2021, doi:10.3390/ijms22020915_

Round 1

Reviewer 1 Report

The authors present the current research in the field of microbiome and ischemic stroke. The topic itself is very interesting and of high clinical importance.

I have to admit that as a stroke interventionalist, I have very shallow knowledge of the topic of microbiome and ischemic stroke. Hence, I can basically rate the manuscript for readability but not for scientific content.

Major point: The review presents many studies. This is a good thing as the review is very detailed, but for the interested non-expert reader, the density of information is very high. Some paragraphs felt like study after study was cited, and I would have liked more connection to pathophysiology and more critical reasoning in between. Maybe a more hypothesis-driven or pathophysiology-driven presentation of results would make the review easier to comprehend for the non-expert reader.

Minor points:

- Section 2 was very long and I somehow lost focus during reading, results should be either structured by species, by bacterial strain or by pathophysiological mechanism
- Section 4 seemed a little bit out of context to me (neurodegeneration and behaviour is not in the focus of ischemic stroke)
- Figure 1 currently shows a unidirectional arrow (how gut affects stroke) for most factors, but the presented literature suggests a bidirectional process (e.g. stroke triggers inflammation)

- "gastrointestinal complications after stroke affect poor patient treatment results, including delayed outcome, increased mortality, and progressive neurological deficits." Reference missing
- "It is believed that the digestive tract is the main organ of the immune response, which has been equipped with the largest pool of immune cells, accounting for over 70% of the entire immune system [15]." Reference 15 does not provide any data or reference to this statement of "over 70%" please cite another reference
- "In particular, the question arises, what is the driving force behind the translocation of intestinal bacteria, is the displaced strain specific or dependent on general intestinal diversity, and is it specific for the colonized organ?" I did not understand why this question is so prominent in the conclusion. I did not see the topic of translation of intestinal bacteria to other organs (e.g. the lung) as a strong focus of the review.

Author Response

The changes proposed by the reviewers are marked in red. Changes proposed by the editorial office are highlighted in green. Reviewer # 1.The authors present the current research in the field of microbiome and ischemic stroke. The topic itself is very interesting and of high clinical importance. Thanks. I have to admit that as a stroke interventionalist, I have very shallow knowledge of the topic of microbiome and ischemic stroke. Hence, I can basically rate the manuscript for readability but not for scientific content. Thanks. Major point: The review presents many studies. This is a good thing as the review is very detailed, but for the interested non-expert reader, the density of information is very high. Some paragraphs felt like study after study was cited, and I would have liked more connection to pathophysiology and more critical reasoning in between. Maybe a more hypothesis-driven or pathophysiology-driven presentation of results would make the review easier to comprehend for the non-expert reader. This paper summarizes the current state of knowledge on the influence of the gut microbiota on ischemic brain injury and the impact of an ischemic brain event on the gut microbiota. We summarized the results obtained in animal models of cerebral ischemia in mice, rats and monkeys, as well as the results of studies in patients with ischemic stroke. At the same time, we highlight the existing discrepancy between these results (e.g. see section 2) and the need for more information. We had to present it in order to understand the complexity of the phenomena studied and emerging problems. Some possible mechanisms are presented in section 3 and in the updated figure 1. The data presented is difficult to interpret due to conflicting information from animal and human studies (see lines 93-95, 97, 145-146, 150-151, 162-165, 168-169, 229-231, 233-235, 241-242, 244-245) because the studies are in the infancy phase. In addition, the data have not been checked or confirmed by numerous laboratories or clinics. Minor points: - Section 2 was very long and I somehow lost focus during reading, results should be either structured by species, by bacterial strain or by pathophysiological mechanism It is difficult to systematize the data as suggested by the reviewer due to the existing discrepancy between these results in animals and humans and inside animals and humans, as presented in section 2 lines 93-95, 97, 150-152, 162-165, 168-169. Mechanisms are extremely difficult to propose due to contradictory information or mutually exclusive data, e.g. within the studied groups: animals or humans. - Section 4 seemed a little bit out of context to me (neurodegeneration and behaviour is not in the focus of ischemic stroke) This section was deleted. Thanks. - Figure 1 currently shows a unidirectional arrow (how gut affects stroke) for most factors, but the presented literature suggests a bidirectional process (e.g. stroke triggers inflammation) The figure has been corrected as suggested by the reviewer. Thanks. - "gastrointestinal complications after stroke affect poor patient treatment results, including delayed outcome, increased mortality, and progressive neurological deficits." Reference missing Relevant references were added in lines 77-79. - "It is believed that the digestive tract is the main organ of the immune response, which has been equipped with the largest pool of immune cells, accounting for over 70% of the entire immune system [15]." Reference 15 does not provide any data or reference to this statement of "over 70%" please cite another reference We are presenting original sentence from reference 15. “The gastrointestinal tract is thought to be a major immune organ which was equipped with the largest pool of immune cells, accounting for more than 70% of the entire immune system” (p. 2, left column, first 4 lines). We changed “over” for original “more than” (line 69 in the MS). - "In particular, the question arises, what is the driving force behind the translocation of intestinal bacteria, is the displaced strain specific or dependent on general intestinal diversity, and is it specific for the colonized organ?" I did not understand why this question is so prominent in the conclusion. I did not see the topic of translation of intestinal bacteria to other organs (e.g. the lung) as a strong focus of the review. This sentence was deleted from the conclusion section. Thanks.

Reviewer 2 Report

In the manuscript 'The role of gut microbiota in ischemic stroke’, the authors conducted a brief literature review about ischemic stroke and changes in gut microbiota in humans and some animal studies. The manuscript describes how studies about stroke and gut microbiota are pivotal to gain insights into the mechanism of action. However, the reviewer is concerned that the manuscript lacks conceptual innovation about new areas of stroke and microbiota research and mechanism of action of gut microbiota.

Major Comments:

1)   The role of microbiota in stroke appears to be associative from the manuscript in its current form therefore it is unclear to the reviewer whether disrupted gut microbiota could cause a stroke or stroke causes gut microbiota changes that lead to subsequent stroke-associated morbidities.

2)   Please refer to the actual landmark or research studies from both humans and animal models in the introduction and other sections, and discuss the similarities and contrasts both in terms of gut microbiota and stroke-related effects rather than citing a review paper. Also, it appears that there is a redundancy in the introductory part at certain sections, for instance, section 5 (lines 219-229).

3)   Please provide rationale on why authors only focused on limited mice or rat studies and not provide findings from additional models of spontaneous stroke or middle cerebral artery occlusion. There have been multiple studies from different groups showing 1) how diet quality affects microbiota that can lead to stroke, 2) how there could be some inherited strains of bacteria that are associated with hypertension or stroke, and 3) how bacterial metabolites could promote or worsen metabolism/hypertension that can lead to early stroke onset.

4)   Please provide a discussion on the about 1) new areas of dietary interventions and microbiota research, for instance, worse effects of poor diet (high-fat) or benefits of good diet (high fiber or high protein) that improve cardiovascular control, weight management and many more other risk factors for stroke and 2) mechanism of action of gut microbiota such as improving metabolism or hDAC or mMP-9 inhibition.

5)   In section 4 on intestinal microbiota versus behavior changes, the link between stroke and other neurological disorders is unclear and thus requires further explanation.

6)   Please provide a brief discussion about the 1) factors that should be taken into consideration when analyzing future clinical trials on microbiota and stroke, and 2) key questions that are still unanswered and should be targeted in the future for a better understanding of how gut microbiota might impact stroke prevention or post-stroke recovery.

Author Response

The changes proposed by the reviewers are marked in red. Changes proposed by the editorial office are highlighted in green. Reviewer #2. In the manuscript 'The role of gut microbiota in ischemic stroke’, the authors conducted a brief literature review about ischemic stroke and changes in gut microbiota in humans and some animal studies. The manuscript describes how studies about stroke and gut microbiota are pivotal to gain insights into the mechanism of action. However, the reviewer is concerned that the manuscript lacks conceptual innovation about new areas of stroke and microbiota research and mechanism of action of gut microbiota. Please consider that in this paper, we present the mutual interaction between the gut microbiota and stroke as well as stroke and gut microbiota (see lines 17-20, 62-63, 65-66, 74-75, 92-93). This interaction is clearly shown now in the modified figure 1 and partially in section 3. In figure 1, we suggest a possible interaction or possible pathophysiology without detailed mechanisms that are currently unknown due to the lack of data (see lines 81-82). It is difficult to comment on the interplay mechanisms at present, as research on this subject is sparse in animals and very limited in the clinical conditions, generally studies are in the infant stage. Data found in the literature and presented in this manuscript are often contradictory or mutually exclusive (see lines 93-95, 97, 145-146, 150-151, 162-165, 168-169, 229-231, 233-235, 241-242, 244-245). In addition, the data have not been verified or confirmed by numerous laboratories or clinics. In summary, there is no solid basis for speculative hypotheses or mechanisms. Major Comments: 1) The role of microbiota in stroke appears to be associative from the manuscript in its current form therefore it is unclear to the reviewer whether disrupted gut microbiota could cause a stroke or stroke causes gut microbiota changes that lead to subsequent stroke-associated morbidities. In the manuscript, we present the latest results of animal and human studies on two-way communication along the gut-brain-gut axis in ischemic stroke (e.g. see the updated figure 1). This paper summarizes the current state of knowledge on the influence of the gut microbiota on ischemic brain injury and the impact of an ischemic brain event on the gut microbiota. We summarized the results obtained in animal models of cerebral ischemia in mice, rats and monkeys, as well as the results of studies in patients with ischemic stroke. 2) Please refer to the actual landmark or research studies from both humans and animal models in the introduction and other sections, and discuss the similarities and contrasts both in terms of gut microbiota and stroke-related effects rather than citing a review paper. Also, it appears that there is a redundancy in the introductory part at certain sections, for instance, section 5 (lines 219-229). Mainly in sections 2 and 4 we present both the similarities and contrasts in animal and human research and additionally, the contrasts inside animal and human research (see lines 93-95, 97, 145-146, 150-151, 162-165, 168-169, 229-231, 233-235, 241-242, 244-245). We moved the introductory part of section 5 to introduction lines 60-67, and now section 5 is section 4. 3) Please provide rationale on why authors only focused on limited mice or rat studies and not provide findings from additional models of spontaneous stroke or middle cerebral artery occlusion. There have been multiple studies from different groups showing 1) how diet quality affects microbiota that can lead to stroke, 2) how there could be some inherited strains of bacteria that are associated with hypertension or stroke, and 3) how bacterial metabolites could promote or worsen metabolism/hypertension that can lead to early stroke onset. We focused on the currently available studies on mice, rats, monkeys (see lines 108, 111, 122) and humans linking gut microbiota to stroke and the impact of stroke on changes in gut microbiota. The topic of the work was not the influence of diet on gut microbiota or hypertension and the influence of hypertension on stroke. In the manuscript, the aim was to present the latest results of animal and human studies on two-way communication along the gut-brain-gut axis in ischemic stroke. 4) Please provide a discussion on the about 1) new areas of dietary interventions and microbiota research, for instance, worse effects of poor diet (high-fat) or benefits of good diet (high fiber or high protein) that improve cardiovascular control, weight management and many more other risk factors for stroke and 2) mechanism of action of gut microbiota such as improving metabolism or hDAC or mMP-9 inhibition. The topic of the work was not devoted to the influence of diet on microbiota research and mechanism of action of microflora on hDAC and mMP-9. Actually, the aim was to present the latest results of animal and human studies on two-way communication along the gut-brain-gut axis in ischemic stroke. In the manuscript, we would like to present the latest results of animal and human studies on two-way communication along the gut-brain-gut axis in ischemic stroke. 5) In section 4 on intestinal microbiota versus behavior changes, the link between stroke and other neurological disorders is unclear and thus requires further explanation. This section has now been removed from the manuscript due to taking out of context. 6) Please provide a brief discussion about the 1) factors that should be taken into consideration when analyzing future clinical trials on microbiota and stroke, and 2) key questions that are still unanswered and should be targeted in the future for a better understanding of how gut microbiota might impact stroke prevention or post-stroke recovery. Our suggestions may be found in section 6, lines 281-286.

Reviewer 3 Report

The presented manuscript in detail summarizes the current state of knowledge of the influence of gut microbiota on ischemic brain damage and pathophysiological processes following an ischemic episode. The authors summarize the findings obtained in animal experiments as well as the results of tests in patients with various disorders of brain function. At the same time, they emphasize the possible discrepancy between these results and the need to obtain further information. The manuscript is written clearly and concisely and includes very new findings published recently.

Author Response

The changes proposed by the reviewers are marked in red. Changes proposed by the editorial office are highlighted in green. Reviewer #3. The presented manuscript in detail summarizes the current state of knowledge of the influence of gut microbiota on ischemic brain damage and pathophysiological processes following an ischemic episode. The authors summarize the findings obtained in animal experiments as well as the results of tests in patients with various disorders of brain function. At the same time, they emphasize the possible discrepancy between these results and the need to obtain further information. The manuscript is written clearly and concisely and includes very new findings published recently. Thank you for your time to evaluate our manuscript.

Round 2

Reviewer 1 Report

The authors have done a reasonable job in adressing my points. 

Author Response

Thanks.

Reviewer 2 Report

The reviewer appreciated the author's response and effort to improve the manuscript.

Major Comments:

The author's response to the important reviewer suggestions that the topic of the work was not devoted to include broad areas such as the stroke-prone rat model, the influence of diet and hypertension on gut microbiota that subsequently cause ischemic stroke, dampened the reviewer's enthusiasm for the manuscript as the reviewer considers it important to discuss the topic in a more mechanistic way.

The reviewer would appreciate a discussion on the putative mechanisms of action of microflora such as hDAC and mMP-9 as they will add depth to the study.

The reviewer agrees that the aim was to present the latest results of animal and human studies on two-way communication along the gut-brain-gut axis in ischemic stroke. However, the reviewer also likes to emphasize that since the role of the gut-brain axis in stroke is in its infancy research bringing in broader insights of the stroke field, current rodent models, and the areas that are being explored will generate holistic questions for the investigators to focus on in the future.

Author Response

Major Comments: The author's response to the important reviewer suggestions that the topic of the work was not devoted to include broad areas such as the stroke-prone rat model, the influence of diet and hypertension on gut microbiota that subsequently cause ischemic stroke, dampened the reviewer's enthusiasm for the manuscript as the reviewer considers it important to discuss the topic in a more mechanistic way. The reviewer would appreciate a discussion on the putative mechanisms of action of microflora such as hDAC and mMP-9 as they will add depth to the study. Response. Two paragraphs (p. 6-7) were included in the final part of the manuscript and they deal shortly with the suggested putative mechanisms of action of microflora. Anyway, especially data on metalloproteinase 9 are very scarce. Four new references were included: 89-92. The new paragraphs were highlighted in yellow for the convenience of the reviewer. The reviewer agrees that the aim was to present the latest results of animal and human studies on two-way communication along the gut-brain-gut axis in ischemic stroke. However, the reviewer also likes to emphasize that since the role of the gut-brain axis in stroke is in its infancy research bringing in broader insights of the stroke field, current rodent models, and the areas that are being explored will generate holistic questions for the investigators to focus on in the future. Response. The authors deeply appreciate the reviewer’s suggestion on the future research related to the gut-brain axis and ischemic stroke. Actually, we decided to put this suggestions at the end of the manuscript.